# Aetiology, antimicrobial susceptibility patterns and factors associated with bacteriuria among HIV-infected women attending Prevention of Mother-to-Child Transmission (PMTCT) clinic at Bukoba Municipality, Tanzania

**Eustadius Kamugisha Felician**[1,2]*, **Loveness Urio**[3], **Mtebe Majigo**[1], **Said Aboud**[1,4]

**1** Muhimbili University of Health and Allied Sciences, Dar es salaam, Tanzania, **2** Bukoba Regional Referral Hospital, Bukoba, Tanzania, **3** Tanzania Field Epidemiology and Laboratory Training Program, Dar es Salaam, Tanzania, **4** National Institute for Medical Research, Dar es Salaam, Tanzania

* eustadius.felician@yahoo.com

## Abstract

### Background

Bacteriuria is the detection of significant bacteria in urine in the presence or absence of signs and symptoms of urinary tract infection (UTI). Bacteriuria in pregnant and lactating HIV-infected women can cause serious complications to women and fetuses for pregnant women. Due to the importance of bacteriuria, we determined the etiology, antimicrobial susceptibility patterns, and factors associated with bacteriuria in HIV-infected women.

### Methods

We conducted a cross-sectional study from January to April 2022 among HIV-infected women attending the Prevention of Mother-to-Child Transmission (PMTCT) clinic at Bukoba Municipality, Tanzania. Clean-catch midstream urine specimens were collected for culture on MacConkey and blood agars. We used colonial characteristics, Gram staining reactions, and biochemical tests to identify bacteria isolates. Data were collected using a structured questionnaire. We used STATA version 15.0 for analysis. An association with bacteriuria was performed using modified poisson regressions. A p-value ≤0.05 was regarded as statistically significant.

### Results

Of the 290 participants, 66 (22.8%) had significant bacteriuria. The predominant bacteria isolates were *Escherichia coli* 21 (31.8%). Among gram-negative bacteria, 17 (34.0%) were extended-spectrum beta-lactamase producers, and 1 (25.0%) of *Staphylococcus aureus* were Methicillin-resistant. *Escherichia coli* showed a high rate of resistance against trimethoprim-sulfamethoxazole 21 (100%), and amoxicillin clavulanic acid 20 (95.0%).

**Data availability statement:** All relevant data are within the paper and its Supporting Information files.

**Funding:** We received a small fund for data collection from TFELTP The funders had no role in study design, data collection and analysis, the decision to publish, or the preparation of the manuscript.

**Competing interests:** The authors of this research have explicitly stated that there are no existing conflicts of interest related to the publication of their work.

*Staphylococcus aureus* was highly resistant to penicillin 4 (100%) and trimethoprim-sulfamethoxazole 4 (100%). The proportion of multi-drug resistant (MDR) strains was 45 (68.2%).

## Conclusions

The prevalence of bacteriuria in HIV-infected women was relatively high. The pathogens were most resistant to trimethoprim-sulfamethoxazole, penicillin and amoxicillin clavulanic acid and more than two-third were MDR. The findings emphasize that the use of antimicrobial agents should be supported by culture and Antimicrobial Susceptibility Testing (AST) results.

## Introduction

Urinary tract infection (UTI) become the common infection of the urinary tract system among community members and within the hospital environment [1,2]. As the most common infectious disease, UTI has affected approximately 150 million diagnosed cases each year worldwide, according to a study conducted in Iran [3]. Even though all age categories are affected by UTI, pregnant and lactating women [4,5] are at higher risk than men due to anatomical position and short urethra [6]. Generally, pregnant and lactating women infected with HIV have a higher chance of developing UTIs than other women [7]. In addition, the human immunodeficiency virus (HIV) in both pregnant and lactating women have a high chance of acquiring opportunistic infections, including UTI, especially in developing countries with limited health services [6,8,9].

UTI occurs among HIV-infected women; it can be with symptoms or asymptomatic bacteriuria (ASB) [10]. Previous studies have revealed a higher prevalence of bacteriuria among women living tha uninfected women [11,12]. ASB is an occurrence of a bacterial colony count more than or equal to $10^5$CFU/mL from a urine sample collected from individuals with no signs and symptoms of UTIs [13]. Symptomatic bacteriuria is based on an individual's showing signs and symptoms with significant bacteria on urine culture accompanied by frequency and urgency of urination, painful urination, blood in urine, and high level of leucocytes in urine [14]. A previous study conducted in Mwanza, Tanzania shows that 21.4% of HIV-infected pregnant women had UTIs [15]. The study conducted in Nigeria found that 16.6% had ASB compared to 4.7% of individuals with symptomatically significant bacteria in urine for the study conducted in Nigeria [12]. Thus, the high incidence and risk of developing UTI during pregnancy and lactation are related to abnormal anatomical and physiological changes that occur during this period [15]. Also, using trimethoprim-sulfamethoxazole prophylaxis and other antibiotics routinely in this population may raise the possibility of getting bacterial multi-drug resistance (MDR) [16].

The most common bacteria causing UTI are *Pseudomonas aeruginosa, Proteus mirabilis, Klebsiella species, Escherichia coli, Enterobacter species, Enterococci, Citrobacter, Staphylococcus aureus* and Coagulase-negative *staphylococcus* [17]. However, *Escherichia coli* is a gram-negative bacteria that mostly causes UTIs, accounting for most (80–90%) infections [18].

Previous studies in different countries have shown various factors associated with bacteriuria in HIV-infected women with symptomatic and asymptomatic UTIs. For example, physiological and hormonal changes that occur in the course of pregnancy, parity, educational status, occupation of pregnant women, history of UTI, Single marital status, CD4 T-cell count below 350 cells/uL among HIV-infected individuals, increased age, multiparity, sexual activity, history of catheterization, immunodeficiency and lower socioeconomic status, history of UTI, aminoaciduria, anemia and diabetes mellitus [12,15,19–24] were among the risk factors reported by different researchers.

Recently, antimicrobial resistance (AMR) in bacteriuria has been increasing worldwide, and some bacteria are virulent and capable of acquiring MDR. For example, *Escherichia coli* are Gram-negative bacteria that can generate a large spectrum of beta-lactam enzymes, making them resistant to most beta-lactam antibiotics [5]. Rates of AMR vary according to geographic locations, and they are directly proportional to the use and misuse of antimicrobials. Antimicrobial therapy for pregnant women is a serious concern during pregnancy [5,25]. Different interventions have been put in place in Tanzania through the Ministry of Health to address the problem of AMR, including establishing antimicrobial stewardship programs, AMR surveillance systems, and strengthening medicines and therapeutic committee and quality improvement teams [26,27]. Data are scarce on the etiology of bacteriuria, drug susceptibility patterns, and associated factors among HIV-infected women. Therefore, the study determined the etiology of bacteriuria, antimicrobial susceptibility patterns, and associated factors among HIV-infected women attending the Prevention of Mother-to-Child Transmission (PMTCT) clinic in Bukoba municipality, Tanzania.

## Materials and methods

### Study design and setting

We conducted a cross-sectional study from 04th January to 30th April 2022 at PMTCT clinics in Bukoba Municipality located in the Kagera region in four high-volume facilities, namely Bukoba Regional Referral Hospital (BRRH), Zamzam Health Centre, Rwamishenye Health Centre, and Kashai Dispensary. Kagera is among the top ten regions with a high prevalence of HIV infection, especially in women. The selected five facilities provide PMTCT services. Each facility attends not less than 20 HIV-infected pregnant women for PMTCT health care services per week. HIV-infected women attending PMTCT clinics who consented to participate in the study were enrolled. We excluded individuals known to have urinary tract abnormalities.

### Sample size and sampling technique

The sample size was 290 HIV-infected women attending PMTCT clinics determined using the Kish and Leslie formula considering a prevalence of 21.4% among HIV-infected pregnant women attending PMTCT in Mwanza at a 95% confidence interval [21]. In addition, a non-probability convenient sampling technique was used to select study participants. We included individuals most accessible at the study site based on daily attendance at PMTCT clinics and who agreed to participate after being eligible.

### Data collection

A pretested structured questionnaire was used to obtain socio-demographic, clinical, and laboratory information. The information included the history of UTI, history of catheterization, educational status, gestational age, parity, gravidity, age, occupational, marital status (single, married and others which included divorced, widows and cohabiting). The hemoglobin level (normal ≥ 11g/dl; moderate: 9.0-10.9g/dl; mild: 7.0-8.9g/dl; severe < 7.0g/dl) were tested during data collection using Haemoglobinometer HemoCue 201 + machine. The CD4 T-cell count were determined using BD FACSPresto™ machine. The CD4 results were extracted from the participants' file and clinic card based on current CD4 cells and the first CD4 results when they started ART initiation. Other information included current viral load result recorded from participant file, PMTCT status, HIV clinical stage, ART regimen, duration of ART use and history of hospitalization.

## Urine specimen collection and handling

Study participants were instructed to collect 10 mL of freshly voided clear-catch midstream urine using pre-labeled (date, identification code, age), leak-proof, wide mouth, sterile, screw-capped plastic container [27]. Participants were informed to clean their hands with water and soap [28] and then cleanse their periurethral area with a sterile cotton swab soaked in normal saline to reduce contamination. Collected specimens were stored in a cold box, transported to the BRRH laboratory, and processed within two hours. In case of unavoidable delay, specimens were refrigerated at 2 - 8°C until time for processing.

## Urine culture and bacteria identification

Urine samples were processed immediately upon arrival at the testing laboratory using the quantitative culture method. The samples were inoculated into sheep blood agar (BA) and MacConkey agar plates (OXOID, Hampshire, United Kingdom) using a sterilized calibrated (0.001mL) wire loop [21]. Incubation of culture plates was done under the presence of oxygen at 37°C for 18-24 hours. Verification of significant growth was done by counting colonies [28]. The bacterial growth of $\geq 10^5$ CFU/mL was considered significant bacteriuria. Specimens that made less than $10^5$ CFU/mL were considered contamination [15]. Following the laboratory standard operating procedures, we identified isolates using conventional methods (colony characteristics, gram staining, and biochemical tests). Gram-negative bacteria were identified using citrate utilization test, oxidase test, urease test, Kligler Iron Agar (KIA), indole production, motility, and hydrogen sulfide production through Sulfide Indole Motility (SIM) test [29]. Catalase and coagulase tests were used to identify gram-positive bacteria.

## Antimicrobial susceptibility testing

Antimicrobial susceptibility testing (AST) was performed using the Kirby–Bauer disk diffusion method recommended by the Clinical and Laboratory Standards Institute (CLSI) [30]. Pure culture colonies of 24-hour growth were suspended in a tube with 4mL of physiological saline to get bacterial inoculums equivalent to 0.5 McFarland turbidity standards. A sterile cotton swab was dipped, rotated across the tube wall to avoid excess fluid, and inoculated on Muller-Hinton agar (Conda Ltd, USA). Then, the antibiotic discs were placed on MHA plates. We selected antimicrobial agents based on the CLSI recommendations and frequent local prescriptions for treating UTIs. The antimicrobials (Oxoid Ltd., UK) for Gram-positive bacteria were penicillin (10 units), erythromycin (15 µg), trimethoprim-sulfamethoxazole (1.25/23.75 µg), tetracycline (30 µg), nitrofurantoin (300 µg) and ciprofloxacin (5 µg). For Gram-negative bacteria, we used ampicillin (10 µg), amoxicillin-clavulanate (20/10 µg), piperacillin-tazobactam (100/10 µg), trimethoprim-sulfamethoxazole (1.25/23.75 µg), nitrofurantoin (300 µg), gentamicin (10 µg), ciprofloxacin (5 µg), ceftriaxone (30 µg), ceftazidime (30 µg), and meropenem (10 µg). The plates were incubated at 37°C for 18-24 hours. The zone of inhibition was measured to the nearest millimeter and interpreted as sensitive, intermediate, or resistance based on the interpretative criteria set by the CLSI [30]. Bacterial isolates resistant to three or more antimicrobial agents in different structural classes were classified as multi-drug resistant (MDR).

## Extended-spectrum beta-lactamase detection

Preliminary identification of extended-spectrum beta-lactamase (ESBL) was conducted by measuring the diameter of zones of inhibition produced for ceftriaxone (30 µg),

ceftazidime (30 μg), or cefotaxime (30 μg) [21,31]. The cutoff point demonstrating a suspicion of ESBL production were: ceftriaxone ≤ 25mm, ceftazidime (30 μg) ≤22mm, and cefotaxime ≤27mm (21). We used the double-disc synergy method to confirm the phenotypic detection of ESBL production [30]. The tested bacteria were fondled onto a Mueller–Hinton agar plate using the same methods for antimicrobial susceptibility testing. Discs of ceftazidime (30 μg), cefotaxime (30 μg), and amoxicillin-clavulanic acid (20/10 μg) were placed 20mm apart on the plate in a straight line, with amoxicillin-clavulanic acid (20/10 μg) in the middle. Any increase in the inhibition zone towards the disk of amoxicillin–clavulanic acid was considered a positive result for ESBL production after incubation at 37°C for 24 hours [30].

## Methicillin-resistant staphylococcus aureus

Methicillin-resistant *Staphylococcus aureus* (MRSA) was confirmed using a cefoxitin disc (30 μg), and strains showing a zone of inhibition of ≤ 21 mm were regarded as MRSA [32,33–36].

## Quality assurance

We performed internal quality control checks to generate reliable data during and after data collection. The entire questionnaire was checked for completeness during and after data collection. All laboratory processes were done following SOPs to ensure quality. Sterility and performance testing for culture media were performed. Reference strains of *S. aureus* (ATCC 25,923), *E. coli* (ATCC 25,922), *E. faecalis* (ATCC29,212), and *P. aeruginosa* (ATCC 27,853) were used as quality control for culture and susceptibility testing throughout the study. Moreover, *E. coli* (ATCC 25,922) and *K. pneumoniae* (ATCC 700,603) reference strains were used as quality control for ESBL detection [37]. All reference strains were obtained from the National Public Health Laboratory (NPHL).

## Data management and analysis

The data were entered in STATA version 15. After the data entry, two generic data verification strategies were employed to verify the precision of the data entry. First, 10% of randomly selected questionnaires were thoroughly checked. Second, descriptive statistics and frequency distributions of each variable were examined. Then, cross-tabulations were carried out to search for additional data entry problems. Before analysis, the data was explored to check for missing values, duplications, and unusual observations. Descriptive statistics were summarized using frequency and proportions for categorical variables, whereas continuous variables were summarized with the measure of central tendency with respective measures of dispersion. Modified Poisson regression analysis was used to determine factors associated with bacteriuria in bivariate analysis with the crude prevalence ratios and their corresponding 95% confidence interval. To account for confounding effects, a multivariable modified Poisson regression analysis was conducted to obtain the adjusted prevalence ratio and 95% confidence interval for bacteriuria.

In bivariate analysis, independent variables that were most shown to be independently associated with bacteriuria of the previously published studies, together with other variables with a p-value of ≤ 0.25, were entered into a model one after a time in the multivariable modified Poisson regression model to determine their contribution to the outcome of interest. We regarded a variable as a confounder if the estimate changed from crude to adjusted by ≥ 10%. The model used a p-value ≤0.05 to declare statistical significance. The parsimonious model was selected based on the model with the lowest Akaike information criteria (AIC).

## Ethical consideration

The Muhimbili University of Health and Allied Sciences (MUHAS) Research and Ethics Committee approved the study with IRB number DA.282/298/06/C/767. Permission for data collection was obtained from BRRH and Bukoba Municipality for four Health Centers. Before data collection, written informed consent was obtained from each participant. Participants enrolled in the study were informed about the study objectives, expected outcomes, benefits, and associated risks. For participants under 18 years, consent was sought from the parent/ guardian and requested assent to the study. All information collected from the study participants was kept confidential. The study did not include details directly identifying participants, such as their names. Only a participant identification number was used in the study. Results of antimicrobial susceptibility testing were communicated on time to the attending doctors/ nurses for management. The study was conducted per the Declaration of Helsinki.

## Results

### Socio-demographic and clinical characteristics of the study participants

A total of 290 HIV-infected women were enrolled in the study. The participants' mean age (±SD) was 29.0 (±5.7) years. Most participants, 233 (80.3%), were from lower health facilities. Of 290 participants, 138 (47.6%) were pregnant, and 152 (52.4%) were lactating women. More than two-thirds, 195 (67.2%), had no signs and symptoms of UTI. The majority, 231 (79.7%), were from urban areas, 205 (70.7%) had a low level of education, and 228 (78.6%) were married. 197 (67.9%) women were employed, and 205 (70.7%) were multipara. The majority of participants, 208 (71.7%), had a baseline CD4 T-cell count of ≥ 350 cells/ μL, and 252 (86.9%) had a current CD4 T-cell count of ≥ 350 cells/ μL. Additionally, the median ART duration was 45 (IQR 1-196), and the median hemoglobin level of the participants was 22 (IQR 2-67) g/dl. Furthermore, 286 (98.6%) participants were using the first-line ART regimen, and 248 (85.5%) had their current viral load status with target not detected (TND), with the majority of 183 (63.1%) being on HIV disease stage one. Additionally, 105 (76.1%) were multigravida, and 63 (45.7%) was in the second trimester (Tables 1 and 2).

### Prevalence of bacteria profile causing significant bacteriuria

Of the 290 study participants, 66 (22.8%) had significant bacteriuria. Almost all 65 (98.5%) of the infected pregnant women had a single infection while only one (1.5%) was dually infected with *E. coli* and *Serratia species* which makes the total number of bacterial isolated to be 66. Gram-negative bacteria (GNB) isolated were more prevalent (n = 50/66: 75.8%) than Gram-positive bacteria (n = 16/66; 24.2%). The most commonly isolated bacteria were *E coli* (n = 21; 31.8%) and *unidentified* GNB were 13 (19.7%) followed by Coagulase- Negative Staphylococcus (CoNS) (n = 10; 15.2%), *P vulgaris* (n = 6; 9.1%), *S aureus* (n = 4; 6.1%) (Table 2). Further, among GNB isolated, 17 (34.0%) were ESBL producers while *E. coli* accounted for 8 (47.0%) of all ESBL producers. Furthermore, among *S. aureus* 1 (25.0%) was MRSA.

### Antimicrobial resistance of isolated bacteria

Table 3 presents the AMR results of Gram-negative bacteria. E. *coli* showed high rate of resistance against trimethoprim-sulfamethoxazole 21 (100%), amoxicillin/clavulanic acid 20 (95.0%) and ampicillin 16 (76.0%) and low rate of resistance 1 (5.0%) for meropenem, 3 (14.0%) for piperacillin-tazobactam, and 3 (14.0%) for nitrofurantoin (Table 3). P. *vulgaris* showed a high rate of resistance to trimethoprim-sulfamethoxazole 6 (100%),

**Table 1. Socio-demographic and clinical characteristics of HIV-infected women attending PMTCT (N = 290).**

|  | Variables | Frequency | Percentage |
|---|---|---|---|
| **Name of facility** | Bukoba RRH | 57 | 19.7 |
|  | Lower HF | 233 | 80.3 |
| **Age (years)** | Mean age (years)* | 29.0 ± 5.7 |  |
|  | ≤ 34 | 235 | 81.0 |
|  | ≥ 35 | 55 | 19.0 |
| **Residence** | Rural | 59 | 20.3 |
|  | Urban | 231 | 79.7 |
| **Occupation** | Employed | 197 | 67.9 |
|  | Unemployed | 93 | 32.1 |
| **Level of education** | Low level | 205 | 70.7 |
|  | Higher level | 85 | 29.3 |
| **Marital status** | Married | 228 | 78.6 |
|  | Unmarried | 62 | 21.4 |
| **PMTCT status** | Pregnant | 138 | 47.6 |
|  | Lactating | 152 | 52.4 |
| **Gravidity (N = 138)** | Primigravida | 33 | 23.9 |
|  | Multigravida | 105 | 76.1 |
|  | Multipara | 205 | 70.7 |
| **Gestational period (N = 138)** | Mean (SD) | 24 (8) |  |
|  | 1st trimester | 19 | 13.8 |
|  | 2nd trimester | 63 | 45.7 |
|  | 3rd trimester | 56 | 40.6 |
| **Current symptoms of UTI** | Yes | 95 | 32.8 |
|  | No | 195 | 67.2 |
| **History of UTI** | Yes | 170 | 58.6 |
|  | No | 120 | 41.4 |
| **History of catheterization** | Yes | 78 | 26.9 |
|  | No | 212 | 73.1 |
| **History of hospitalization (Last 3 months)** | Yes | 13 | 4.5 |
|  | No | 277 | 95.5 |
| **History of diabetes mellitus** | Yes | 6 | 2.1 |
|  | No | 284 | 97.9 |
| **Baseline CD4 T-cell count (cells/ µL)** | ≥350 | 208 | 71.7 |
|  | <350 | 82 | 28.3 |
| **Current CD4 T-cell count (cells/ µL)** | ≥350 | 252 | 86.9 |
|  | <350 | 38 | 13.1 |
| **Duration of ART use (months)** | Median (IQR) | 45 (1-196) |  |
|  | ≤12 | 41 | 14.1 |
|  | 13-24 | 46 | 15.9 |
|  | ≥25 | 203 | 70.0 |
| **Type of ART regime** | First-line | 286 | 98.6 |
|  | Second line | 4 | 1.4 |
| **SP use (N = 138)** | Yes | 78 | 56.5 |
|  | No | 60 | 43.5 |
| **Current viral load status (copies/mL)** | ≥1,000 | 1 | 0.4 |
|  | <200 | 41 | 14.1 |
|  | Target not detected | 248 | 85.5 |

*(Continued)*

**Table 1.** (Continued)

| | Variables | Frequency | Percentage |
|---|---|---|---|
| **HIV disease stage** | Stage I | 183 | 63.1 |
| | Stage II | 74 | 25.5 |
| | Stage III | 28 | 9.7 |
| | Stage III | 5 | 1.7 |
| **Hemoglobin (g/dL)** | Mean (SD) | 22 (2-67) | |
| | Normal | 232 | 80.0 |
| | Moderate | 24 | 8.3 |
| | Mild | 15 | 5.2 |
| | Severe | 19 | 6.5 |

**Table 2.** Distribution of bacteria isolated among HIV-infected women attending PMTCT (N = 66) clinic.

| Bacteria isolated | Frequency | Percentage |
|---|---|---|
| *E. coli* | 21 | 31.8 |
| Unidentified GNB | 13 | 19.7 |
| CoNS | 10 | 15.2 |
| *P. vulgaris* | 6 | 9.1 |
| *S. aureus* | 4 | 6.2 |
| *Acinetobacter spp* | 2 | 3.0 |
| *Klebsiella spp* | 2 | 3.0 |
| *Shigella spp* | 2 | 3.0 |
| *Streptococcus spp* | 2 | 3.0 |
| *Citrobacter spp* | 1 | 1.5 |
| *Enterobacter Spp* | 1 | 1.5 |
| *Pseudomonas aeruginosa* | 1 | 1.5 |
| Serratia Spp | 1 | 1.5 |
| **Total** | **66** | **100** |

GNB=Gram Negative Bacteria, Spp=Species, CoNS= Coagulase-negative staphylococcus.

amoxicillin/clavulanic acid 6 (100%) and ampicillin 4 (66.7%) while sensitivity of 100% to piperacillin-tazobactam, meropenem and nitrofurantoin were observed (Table 3). Table 4 presents the AMR results of Gram-positive bacteria. CoNs showed a high level of resistance to penicillin 9 (90.0%) and trimethoprim-sulfamethoxazole 9 (90.0%) and low resistance to nitrofurantoin 1 (10.0%). Moreover, S. *aureus* was resistant to penicillin 4 (100%) and trimethoprim-sulfamethoxazole 4 (100%) and was sensitive 100% to nitrofurantoin and low resistance to ciprofloxacin, erythromycin, clindamycin and tetracycline by 1 (25%). (Table 4).

## MDR pathogens isolated from HIV-infected women

The proportion of MDR was 45 (68.2%). Gram-negative and Gram-positive bacteria showed 37 (74.0%) and 8 (50.0%) MDR rates, respectively. *E. coli* showed a high rate of MDR among gram-negative bacteria, with 15 (71.4%), whereas *S. aureus* showed an MDR of 3 (75.0%) among gram-positive bacteria. MDR were also found to be high in CoNS 5 (50.0%), *P. vulgaris* 4 (66.7%) and unidentified GNB 10 (76.9%). (Table 5).

**Table 3. Antimicrobial resistance of Gram-Negative bacteria (n = 50) isolated from urine culture among HIV-infected women attending PMTCT at Bukoba Municipality.**

| Isolates (n) | Antimicrobial agents, N (%) | | | | | | | | | | |
|---|---|---|---|---|---|---|---|---|---|---|---|
| | AMP | TZP | AMC | MEM | CN | CIP | F | CTX | CRO | CAZ | CTM/SXT |
| *E, coli* n = 21 | 16 (76) | 3 (14) | 20 (95) | 1 (5) | 5 (24) | 4 (19) | 3 (14) | 10 (48) | 9 (43) | 9 (43) | 21 (100) |
| Unidentified GNB, n = 13 | 9 (69) | 0 (0.0) | 12 (92.2) | 0 (0.0) | 3 (23) | 4 (31) | 3 (23) | 6 (46) | 6 (46) | 5 (39) | 12 (92) |
| *P. vulgaris,* n = 6 | 4 (66.7) | 0 (0.0) | 6 (100) | 0 (0.0) | 1 (17) | 1 (17) | 0 (0.0) | 2 (34) | 1 (17) | 2 (33) | 6 (100) |
| *Acinetobacter* Spp n = 2 | NA | 0 (0.0) | NA | 0 (0.0) | 1 (50) | 1 (50) | NA | 0 (0.0) | 0 (0.0) | 0 (0.0) | 1 (50) |
| *Shigella* spp n = 2 | 2 (100) | 0 (0.0) | 1 (50) | 0 (0.0) | 0 (0.0) | 0 (0.0) | 0 (0.0) | 0 (0.0) | 0 (0.0) | 0 (0.0) | 2 (100) |
| *Klebsiella* spp n = 2 | 2 (100) | 1 (50) | 2 (100) | 0 (0.0) | 2 (100) | 0 (0.0) | 2 (100) | 1 (50) | 1 (50) | 1 (50) | 2 (100) |
| *Citrobacter* n = 1 | 1 (100) | 0 (0.0) | 1 (100) | 0 (0.0) | 1 (100) | 0 (0.0) | 1 (100) | 1 (100) | 1 (100) | 1 (100) | 1 (100) |
| *Enterobacter* spp n = 1 | 1 (100) | 0 (0.0) | 1 (100) | 0 (0.0) | 0 (0.0) | 0 (0.0) | 0 (0.0) | 0 (0.0) | 0 (0.0) | 0 (0.0) | 1 (100) |
| *P aeruginosa* n = 1 | NA | 0 (0.0) | NA | 0 (0.0) | 0 (0.0) | 1 (100) | NA | NA | NA | 0 (0.0) | NA |
| *Serratia* Spp n = 1 | 1 (100) | 0 (0.0) | 1 (100) | 0 (0.0) | 0 (0.0) | 0 (0.0) | 1 (100) | 0 (0.0) | 0 (0.0) | 0 (0.0) | 1 (100) |
| **Total, N = 50** | **36 (72)** | **7 (14)** | **44 (88)** | **1 (2)** | **13 (26)** | **11 (22)** | **10 (20)** | **20 (40)** | **18 (36)** | **18 (36)** | **47 (94)** |

**AMP**: Ampicillin; **TZP**: Piperacillin-tazobactam; **AMC**: Amoxicillin/Clavulanic acid; **MEM**: Meropenem; **CN**: Gentamicin; **CIP**: Ciprofloxacin; **F**: Nitrofurantoin; **CTX**: Cefotaxime; **CRO**: Ceftriaxone; **CAZ**: Ceftazidime and **CTM/SXT**: trimethoprim-sulfamethoxazole; **NA:** Not applicable.

**Table 4. Antimicrobial resistance of Gram-Positive bacteria (n = 16) isolated from urine culture among HIV-infected women attending PMTCT (n = 290) at Bukoba Municipality.**

| Isolates (n) | (CTM/SXT) | F | CIP | PG | E | CD | TE |
|---|---|---|---|---|---|---|---|
| CoNS (n = 10) | 9 (90.0) | 1 (10.0) | 5 (50.0) | 9 (90.0) | 7 (70.0) | 2 (20.0) | 3 (30.0) |
| *S. aureus* (n = 4) | 4 (100.0) | 0 (0.0) | 1 (25) | 4 (100.0) | 2 (50.0) | 1 (25.0) | 1 (25.0) |
| *Streptococcus* Spp (n = 2) | 1 (50.0) | 0 (0.0) | 0 (0.0) | 1 (50.0) | 0 (0.0) | 0 (0.0) | 0 (0.0) |
| **Total n = 16** | **14 (87.2)** | **1 (6.3)** | **6 (37.0)** | **14 (87.5)** | **9 (56.3)** | **3 (18.8)** | **4 (25.0)** |

CoNS: Coagulase-Negative Staphylococcus; CTM/SXT: trimethoprim-sulfamethoxazole; F: Nitrofurantoin; CIP: Ciprofloxacin: PG: Penicillin; E: Erythromycin; CD: Clindamycin and TE: Tetracycline.

### Factors associated with bacteriuria

Bivariate and multivariate analysis showed that no single variables were significantly associated with bacteriuria among HIV-infected women (Table 6).

## Discussion

The prevalence of bacteriuria in HIV-infected women attending PMTCT clinic in Bukoba Municipality was 22.8%. Gram-negative bacteria were the most resistant against trimethoprim-sulfamethoxazole, ampicillin, and amoxicillin/clavulanic acid. Gram-positive bacteria showed a high level of resistance against penicillin and trimethoprim-sulfamethoxazole. The proportion of MDR pathogens was 68.2%. We also noted that there was no factor significantly associated with bacteriuria [10,38,39].

The prevalence of bacteriuria in the current study is higher compared to the findings in the previous studies conducted in Iran 15.6% [31], Iran 13.1% [33], Ethiopia 14.4% [34], and Tanzania 14.6% [35]. The prevalence of bacteriuria in this study was slightly higher than the 21.4% observed in HIV-infected pregnant women in Mwanza five years ago [21] and higher than 15% for the general obstetric population in Mwanza region, Tanzania, reported thirteen years ago [35,40,41]. The difference in the prevalence observed may be due to the small

**Table 5. MDR patterns of bacteria isolated (n = 66) from HIV-infected women attending PMTCT (N = 290) clinic at Bukoba Municipality.**

| Isolate | Total (%) | R1 | R2 | R3 | R4 | ≥R5 | MDR (%) |
|---|---|---|---|---|---|---|---|
| **Gram-negative bacteria** | **50 (75.8)** | **5 (10.0)** | **8 (16.0)** | **18 (36.0)** | **4 (8.0)** | **15 (30.0)** | **37 (74.0)[a]** |
| *E. coli* | 21 (31.8) | 3 (14.3) | 3 (14.3) | 5 (23.8) | 1 (4.8) | 9 (42.8) | 15 (71.4) |
| Unidentified GNB | 13 (19.7) | 2 (15.3) | 1 (7.7) | 5 (38.5) | 2 (15.4) | 3 (23.1) | 10 (76.9) |
| *P. vulgaris* | 6 (9.1) | 0 (0.0) | 2 (33.3) | 3 (50.0) | 0 (0.0) | 1 (16.7) | 4 (66.7) |
| *Acinetobacter* spp | 2 (3.0) | 0 (0.0) | 1 (50.0) | 0 (0.0) | 1 (50.0) | 0 (0.0) | 1 (50.0) |
| *Klebsiella* spp | 2 (3.0) | 0 (0.0) | 0 (0.0) | 1 (50.0) | 0 (0.0) | 1 (50.0) | 2 (100.0) |
| *Shigella* spp | 2 (3.0) | 0 (0.0) | 1 (50.0) | 1 (50.0) | 0 (0.0) | 0 (0.0) | 1 (50.0) |
| Others | 4 (6.0) | 0 (0.0) | 0 (0.0) | 3 (75.0) | 0 (0.0) | 1 (25.0) | 4 (100.0) |
| **Gram positive bacteria** | **16 (24.2)** | **2 (12.5)** | **6 (37.5)** | **4 (25.0)** | **1 (6.3)** | **3 (18.8)** | **8 (50.0)[a]** |
| CoNS | 10 (15.2) | 0 (0.0) | 5 (50.0) | 2 (20.0) | 0 (0.0) | 3 (30.0) | 5 (50.0) |
| *S. aureus* | 4 (6.1) | 0 (0.0) | 1 (25.0) | 2 (50.0) | 1 (25.0) | 0 (0.0) | 3 (75.0) |
| *Streptococcus* spp | 2 (3.0) | 2 (100.0) | 0 (0.0) | 0 (0.0) | 0 (0.0) | 0 (0.0) | 0 (0.0) |
| **Total** | **66 (100)** | **7 (10.6)** | **14 (21.2)** | **22 (33.3)** | **5 (7.8)** | **18 (27.3)** | **45 (68.2)[a]** |

Note:

[a]Percent is computed from the total number of isolates, based on which MDR definition is applied. spp: species; GNB: Gram-Negative Bacteria; CoNS: Coagulase-Negative Staphylococcus; R0: no resistance; R1: resistance to one; R2: resistance to two; R3: resistance to three; R4: resistance to four; R5: resistance to five or more antibiotics; MDR: Multidrug-resistant; Others: *Citrobacter* spp, *Enterobacter* spp, P. aeruginosa, *Serratia* spp.

sample size of HIV-infected pregnant and lactating women and non-HIV pregnant and lactating women, which might decrease the overall prevalence when compared with the present study [10,39]. However, the findings were lower than those reported from previous studies conducted in Nepal, which were 30.5% [19], Nigeria, 25.3% [37], and India, 24.0% [38], 28.0% [39]. The variations might be due to differences in geographical location (Europe, Asia, Sub-Saharan) with different climatic conditions. The deployment of diverse methodologies (simple random, systematic, and convenient sampling) might be another factor in variation prevalence. In addition, social behavior of the population, the model of educational status, study settings (primary care, community-based, or in hospitals), sample size of study and time of the study conducted also contribute the difference in prevalence of bacteriuria. Further, the prevalence of bacteriuria in the study is higher compared to the findings, 2%–10%, reported in developed countries [42], and might be due to low socioeconomic status and differences in the level of healthcare delivery [10,43].

The proportion of Gram-negative bacteria identified in this study was higher than that of Gram-positive bacteria (75.8% vs. 24.2%). This finding is in line with studies conducted in Ethiopia [15], Tanzania [44], and Ethiopia [10]. The high prevalence of gram-negative bacteria might be due to its unique structure, which assists in attachment to the uro-epithelium and prevents pathogens from being washed away by urine, as well as allowing for growth and tissue invasion resulting in invasive infection and pyelonephritis in women [45].

*E. coli* was found to be the most prevalent bacteria isolated in the current study. The finding agrees with previously reported at BMC, Mwanza, Muhimbili National Hospital, and Hydom Hospital in Tanzania [21,35,46,47,48]. The findings are also in line with those reported in previous studies in Ethiopia (37.3%) [40], Kenya (38.8%) [41], and Tanzania (64.2%) on the surveillance done on the health facilities [49]. This could be due to virulence factors harbored by the bacteria to colonize the urinary epithelium, the anatomical proximity between the anus, vagina, urethra, and difficulty in maintaining personal hygiene. Another factor in isolating a higher rate of *E. coli* might be the significant abundance of *E. coli* as fecal flora, which, in turn, ascends through genitalia to cause UTI [50]. The numerous virulence

**Table 6. Factors associated with bacteriuria among HIV-infected women (N = 290) attending PMTCT at Bukoba Municipality.**

| Variables | SB Positiven (%) | Bivariate Analysis | | Multivariable Analysis | |
|---|---|---|---|---|---|
| | | Crude PR (95% CI) | P-Value | Adjusted PR (95% CI) | P-Value |
| **Name of facility** | | | | | |
| Bukoba RRH | 14 (24.6) | 1.1 (0.61-1.99) | 0.75 | 1.90 (0.89-4.05) | 0.096 |
| Lower HF | 52 (22.3) | 1 | | | |
| **Age (years)** | | | | | |
| ≤34 | 54 (23.0) | 1 | | | |
| ≥35 | 12 (21.8) | 0.95 (0.51-1.77) | 0.87 | | |
| **Residence** | | | | | |
| Rural | 16 (27.1) | 1 | | | |
| Urban | 50 (21.7) | 0.80 (0.45-1.40) | 0.433 | | |
| **Occupation** | | | | | |
| Employed | 46 (23.4) | 1 | | | |
| Unemployed | 20 (21.5) | 0.92 (0.45-1.56) | 0.759 | | |
| **Level of education** | | | | | |
| Low level of education | 48 (23.4) | 1.1 (0.64-1.90) | 0.716 | | |
| Higher level of education | 18 (21.2) | 1.07 (0.61-1.85) | 0.820 | | |
| **Marital status** | | | | | |
| Married | 49 (21.5) | 1 | | | |
| Unmarried | 17 (27.4) | 1.28 (0.73-2.22) | 0.387 | | |
| **PMTCT status** | | | | | |
| Pregnant | 28 (20.3) | 0.81 (0.50-1.32) | 0.402 | 0.80 (0.48-1.33) | 0.383 |
| Lactating | 38 (25.0) | 1 | | | |
| **Gravidity (N = 138)** | | | | | |
| Primigravida | 6 (18.2) | 1 | | | |
| Multigravida | 22 (21.0) | 1.25 (0.47-2.84) | 0.758 | | |
| **Parity** | | | | | |
| Nullipara | 7 (20.0) | 1 | | | |
| Primipara | 14 (28.0) | 1.4 (0.57-3.47) | 0.467 | | |
| Multipara | 45 (22.0) | 1.1 (0.50-2.43) | 0.819 | | |
| **Gestational period in trimester (N = 138)** | | | | | |
| 1st and 2nd | 17 (21.8) | 1 | | | |
| 3rd | 8 (11.3) | 1.0 (0.72-1.52) | 0.821 | | |
| **Current symptoms of UTI** | | | | | |
| Yes | 22 (23.2) | 1.03 (0.62-1.71) | 0.921 | 1.11 (0.63-1.96) | 0.720 |
| No | 44 (22.6) | 1 | | | |
| **Current CD4 T-cell count (cells/uL)** | | | | | |
| ≥350 | 55 (21.8) | 1 | | | |
| <350 | 11 (29.0) | 1.33 (0.69-2.53) | 0.393 | 1.34 (0.67-2.68) | 0.408 |
| **Type of ART regimen** | | | | | |
| First line | 65 (22.7) | 0.91 (0.13-6.55) | 0.925 | | |
| Second line | 1 (25.0) | 1 | | | |
| **HIV disease stage** | | | | | |
| Stage I&II | 59 (23.0) | 1 | | | |
| Stage III&IV | 7 (21.2) | 0.92 (0.42-2.02) | 0.843 | | |

factors used for colonization and invasion of the urinary epithelium, such as P-fimbriae or pili adherence factors, mediate the attachment of *E. coli* to uroepithelial cells [10].

Unidentified gram-negative bacteria were the second predominant bacterial isolates, with an isolation rate of 19.7%. The findings align with those reported previously in pregnant women in rural Tanzania (32.4%) [46]. CoNS was the third dominant bacterial isolate in the study, with an isolation rate of 15.2%. The results are in agreement with those reported from previous studies conducted in Gondar [15], Bahir Dar [51], Addis Ababa [52,53], Dire Dawa Ethiopia [34], and India [38]. ESBL producers were found to be 34.0%, which is higher than 13.3% for the study conducted in Mwanza [21]. One reasonable explanation might be due to previous exposure to antimicrobial agents, which then select drug-resistant mutants in the digestive tract [10].

Knowledge of AST patterns of bacteriuria pathogens is very important to guide clinicians in selecting and using effective antimicrobial agents to treat patients with UTIs. In this study, the AMR pattern of Gram-negative bacteria demonstrated that the majority of isolates showed resistance to ampicillin (76%), amoxicillin-clavulanic acid (90%) and trimethoprim-sulfamethoxazole (96%), meropenem (2.0%), piperacillin-tazobactam (14%), ciprofloxacin (26%) nitrofurantoin (22%), gentamicin (26%), ceftazidime (36%), ceftriaxone (36%), cefotaxime (42%). Low resistance levels were observed against meropenem, piperacillin-tazobactam, nitrofurantoin, ciprofloxacin, and gentamicin. The findings might be due to possible empirical therapy, particularly in the study area. In the current study, the highest resistance shown to trimethoprim-sulfamethoxazole (96%), amoxicillin-clavulanic acid (90%), and ampicillin (76%) among gram-negative bacteria could be due to irrational use of the antimicrobial drugs over many years, access without prescription and indiscriminate use of the same which could lead to an increased AMR [54,55,56,57]. This aligns with the previously conducted study in Mwanza, Tanzania [21,39] and Ethiopia [34,51].

Among Gram-positive bacteria tested against commonly used antimicrobial agents, AST patterns of the isolates revealed a high level of resistance to penicillin (87.5%) and trimethoprim-sulfamethoxazole (87.5%). The findings can be explained possibly due to easy access without prescription and frequent use of the drugs in the study population. In addition, this might be due to the use of this antibiotic as prophylaxis against opportunistic infections in HIV-infected individuals. The majority of bacterial isolates were sensitive to nitrofurantoin (93.7%), vancomycin (81.2%), clindamycin (81.2%), tetracycline (75.0%), and ciprofloxacin (62.5%). The susceptibility of Gram-positive bacteria to nitrofurantoin, which is relatively higher than that reported from Mwanza, Tanzania [21] with a sensitivity of 83.3%, could be explained by less frequent utilization of nitrofurantoin in the study population.

MDR was seen in 68.2% of the HIV-infected women with bacteriuria. The study findings are lower compared to the study reported previously in Ethiopia 74% [5,52] also, is lower than previous study reports from Bahir Dar 91.3%, Jigjiga 96.0%, Gondar 95.0% (15) Dire Dawa 100.0% both conducted in Ethiopia and Kenya 96.0%. The study indicated that MDR was high among the commonly used antimicrobial agents in the study population. The variations in MDR might be due to repeated, inappropriate, and incorrect use of antimicrobial agents in empirical treatment and poor infection prevention and control strategies, raising the prevalence of resistant pathogens in the study population [10]. In addition, the alarming number of MDR cases might be due to the prescription of antimicrobials without laboratory guidance and irrational administration of antimicrobial agents in empirical treatment. Besides, over-the-counter sales of antimicrobials without prescription is a possible factor for increased AMR [10,39].

Among the factors included in the analysis, no factor was associated with bacteriuria through multivariable modified Poisson regression analysis. This is similar to the study

conducted in Lusaka, Zambia [38]. However, participants with current symptoms of UTI and a CD4 count of less than 350 cells/μl were more likely to have bacteriuria, although they were not significantly associated with bacteriuria. This is similar to the study conducted in Mwanza, Tanzania [21]. Even though there is no clear explanation for these variations, it could be attributable to the differences in cultural practice, hygiene, and norms on sex issues in different areas [58].

The current study had some limitations observed, including limited availability of biochemical identification tests that lead to unidentified Gram-negative bacteria. The findings may not be generalizable to the general population. Participants might experience a recall bias, where some could not remember some of the relevant information during data collection.

## Conclusions

The prevalence of bacteriuria among HIV-infected women was found to be relatively high. *E. coli* was the most prevalent bacteria isolated. Most pathogens showed high resistance to trimethoprim-sulfamethoxazole, amoxicillin/clavulanic acid, and penicillin and low resistance to nitrofurantoin. ESBL producers, MRSA, and MDR strains were high in the study population. There is a need for molecular characterization of ESBL, MRSA, and MDR stains to explore more these resistance patterns. The absence of significant associations with bacteriuria risk factors indicates a need for comprehensive investigations to identify potential predictors and effectively target interventions. The findings emphasize that the use of antimicrobial agents should be supported by culture and AST results. However, appropriate prescription and use of antibiotics are necessary, emphasizing the importance of continuous education for healthcare providers on antibiotic stewardship. Furthermore, incorporating strategies for infection prevention and control of AMR would create a more comprehensive approach to addressing antimicrobial resistance. Finally, nitrofurantoin should continue to be prescribed as it aligns with the Tanzania Standard Treatment Guidelines and National Essential Medicines List (STG/NEMLIT) to avoid irrational antimicrobial therapies.

### Strength of the study

The study population consisted of HIV-infected women from rural and urban areas of Bukoba Municipality. The study was conducted in different levels of healthcare facilities.

## Supporting information

**S1 File. Data availability.**
(XLSX)

## Acknowledgements

We acknowledge FELTP, MUHAS, MoH, Bukoba RRH and Bukoba municipality for accepting us to carry out this study. We also thank all the women who consented to participate in this study. We wishes to acknowledge the staff of Bugando Medical Centre, histopathology laboratory, who participated in the quality assurance of the histology section readings for parasite detection. Finally, we thanks Welington Mosses for technical assistance at Bukoba RRH laboratory.

## Author contributions

**Conceptualization:** Eustadius Kamugisha Felician, Loveness Urio, Mtebe Majigo, Said Aboud.

**Data curation:** Eustadius Kamugisha Felician, Mtebe Majigo.

**Formal analysis:** Eustadius Kamugisha Felician.

**Investigation:** Eustadius Kamugisha Felician.

**Methodology:** Eustadius Kamugisha Felician.

**Project administration:** Eustadius Kamugisha Felician.

**Resources:** Eustadius Kamugisha Felician.

**Software:** Eustadius Kamugisha Felician.

**Supervision:** Loveness Urio, Mtebe Majigo, Said Aboud.

**Validation:** Eustadius Kamugisha Felician.

**Visualization:** Eustadius Kamugisha Felician, Loveness Urio, Mtebe Majigo, Said Aboud.

**Writing – original draft:** Eustadius Kamugisha Felician, Mtebe Majigo, Said Aboud.

**Writing – review & editing:** Eustadius Kamugisha Felician, Said Aboud.

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
