## [Decision Letter · Decision Letter 0]

29 Oct 2024

PONE-D-24-33570Aetiology, antimicrobial susceptibility patterns and factors associated with bacteriuria among HIV-infected women attending prevention of mother to child transmission clinic at Bukoba Municipality, TanzaniaPLOS ONE

Dear Dr. Felician,

Thank you for submitting your manuscript to PLOS ONE. After careful consideration, we feel that it has merit but does not fully meet PLOS ONE’s publication criteria as it currently stands. Therefore, we invite you to submit a revised version of the manuscript that addresses the points raised during the review process.

We look forward to receiving your revised manuscript.

Kind regards,

Mabel Kamweli Aworh, DVM, MPH, PhD. FCVSN

Academic Editor

PLOS ONE

**Journal Requirements:**

https://doi.org/10.2147/TCRM.S267101

https://urologyresearchandpractice.org/content/files/sayilar/173/buyuk/251-260.pdf

In your revision ensure you cite all your sources (including your own works), and quote or rephrase any duplicated text outside the methods section. Further consideration is dependent on these concerns being addressed.

We received a small fund for data collection from TFELTP

4. We note that your Data Availability Statement is currently as follows: All relevant data are within the manuscript and its Supporting Information files

6. PLOS requires an ORCID iD for the corresponding author in Editorial Manager on papers submitted after December 6th, 2016. Please ensure that you have an ORCID iD and that it is validated in Editorial Manager. To do this, go to ‘Update my Information’ (in the upper left-hand corner of the main menu), and click on the Fetch/Validate link next to the ORCID field. This will take you to the ORCID site and allow you to create a new iD or authenticate a pre-existing iD in Editorial Manager.

7. One of the noted authors is a group or consortium. In addition to naming the author group, please list the individual authors and affiliations within this group in the acknowledgments section of your manuscript. Please also indicate clearly a lead author for this group along with a contact email address.

8. Please amend either the abstract on the online submission form (via Edit Submission) or the abstract in the manuscript so that they are identical.

**Additional Editor Comments:**

To further enhance the clarity and quality of your revised manuscript, I kindly request that you address the following additional issues:

When submitting the revised manuscript, please ensure that line numbers and page numbers are included for ease of reference in the review process.In the Introduction section, please define PMTCT upon its first mention and clarify any other abbreviations that have not been previously defined.In the Discussion section, please provide an interpretation of your study results, discussing the implications of these findings in the context of similar studies in the literature. Avoid repeating results in this section.Remove the sub-headings "Limitations" and "Recommendations." The limitations of the study should be detailed in the final paragraph of the Discussion section, while recommendations based on your findings should appear as the last paragraph of the Conclusion section.Ensure that bacterial names are consistently italicized throughout the manuscript. Specifically, please italicize *E. coli* in the Conclusion section.Lastly, please review the manuscript thoroughly to correct any typographical or grammatical errors.

Reviewers' comments:

Reviewer's Responses to Questions

**Comments to the Author**

1. Is the manuscript technically sound, and do the data support the conclusions?

Reviewer #1: Yes

Reviewer #2: Yes

Reviewer #3: Yes

Reviewer #4: Partly

2. Has the statistical analysis been performed appropriately and rigorously? 

Reviewer #1: Yes

Reviewer #2: Yes

Reviewer #3: Yes

Reviewer #4: Yes

3. Have the authors made all data underlying the findings in their manuscript fully available?

Reviewer #1: Yes

Reviewer #2: Yes

Reviewer #3: No

Reviewer #4: Yes

4. Is the manuscript presented in an intelligible fashion and written in standard English?

Reviewer #1: Yes

Reviewer #2: Yes

Reviewer #3: Yes

Reviewer #4: Yes

5. Review Comments to the Author

**Reviewer #1: ** The study offers valuable insights into the high prevalence of bacteriuria and alarming levels of antibiotic resistance among HIV-infected women in a resource-limited setting. The findings align with regional studies; however, the absence of significant associations with bacteriuria risk factors indicates a need for more comprehensive investigations to identify potential predictors and effectively target interventions. The identification of a high proportion of multidrug-resistant (MDR) pathogens (68.2%) raises concern and underscores the urgent need for improved antimicrobial stewardship, rational antibiotic use, and enhanced infection control measures to combat antibiotic resistance in vulnerable populations.

The data and analyses robustly support the claims, and the described methodology is sufficient for reproducing the experiments. The authors appropriately contextualize their claims within existing literature, fairly treating previous studies. They compare the prevalence of bacteriuria and MDR pathogens in their research with findings from similar studies conducted in Iran, Ethiopia, Nepal, Nigeria, and other regions of Tanzania, acknowledging geographical, methodological, and healthcare delivery differences that may explain variations in results.

While the study effectively highlights high resistance rates linked to irrational use and easy access, incorporating clinical data on antibiotic use within and outside the PMTCT clinic could further enhance the analysis. This addition would facilitate correlations between resistance patterns and factors such as age, socioeconomic status, or prior antimicrobial use, leading to a more nuanced understanding of AMR drivers and improving the study's capacity to inform targeted interventions.

The authors' recommendations serve as a good starting point but could be strengthened for greater impact. Emphasizing the importance of continuous education for healthcare providers on antibiotic stewardship would enhance their recommendations. Additionally, defining rational use in the context of local resistance patterns and incorporating strategies for infection prevention and control would create a more comprehensive approach to addressing antimicrobial resistance.

The write-up could also benefit from minor edits to correct organization and typographical errors for improved clarity.

**Reviewer #2:**  1. In the title, instead of “ prevention of mother to child transmission” capitalize the first words of the clinic name “ Prevention of Mother to Child Transmission (PMTCT)”.

2. While the ethical considerations are well addressed, including information about how confidentiality was maintained during the study would strengthen this section.

Good job.

**Reviewer #3: ** GENERAL

Overall, this study is technically sound, but there are some sections where the sentences are difficult to follow. I recommend that the authors collaborate with an English writing coach to enhance the phrasing and improve the readability of the text.

The inclusion of confounding variables in your statistical analysis is commendable.

Revisions Required:

1. The in-text citations are inconsistent (a mix of round and square brackets). Please note that PLOS One requires the use of reference numbers in square brackets, so this needs to be corrected throughout the manuscript.

2. Include page numbers and line numbers in the submitted manuscript for easier reference.

3. Ensure that the spacing between paragraphs is consistent across the document.

4. Under the “Financial Disclosure” section, please write out the full name of the funding body, specifically for TFELTP.

TITLE PAGE

The corresponding author list should not include physical addresses; only email addresses are required. • List the corresponding author’s initials in parentheses after the email address. (See PLOS One Title, Author, Affiliations Formatting Guidelines)

ABSTRACT

1. In the sentence “17 (34.0%) of gram-negative bacteria were Extended Spectrum Beta Lactamase (ESBL) procedures,” please clarify if "procedures" is the intended term, as it seems out of context. Consider revising for clarity.

2. In the phrase “Escherichia coli showed high rate of resistance rate against trimethoprim-sulfamethoxazole 21(100%),” please choose one term for consistency. For example, use "high rate of resistance" or "resistance rate" but not both.

3. The statement under Recommendations is vague. The authors should specify whether they are referring to prescribed or over-the-counter antimicrobial agents.

INTRODUCTION

Paragraph 3: Please state the country in which Mwanza is located for clarity.

Paragraph 4: Ensure that genus/species names are written in italics to adhere to scientific formatting conventions.

MATERIAL AND METHODS

Urine specimen collection and handling: 10 ml not ten mL

Reference 28 is incorrectly cited. This study examined the effect of handwashing with water or soap and did not involve the use of a sterile cotton swab soaked in normal saline to reduce contamination. Therefore, this reference should be placed after "soap" for accurate attribution.

Urine culture and bacteria identification: It would be great if the authors could state why a hydrogen sulfide production test was performed when the Kligler Iron Agar tests for hydrogen sulfide production.

RESULTS

Socio-demographic and clinical characteristics of the study participants:

There is a lack of agreement between the data in Table 2 and the accompanying text (See the last line under this section of the manuscript).

Table 3: Isolates “genus and species name” should be in italics.

The letter case for the acronym for Coagulase Negative Staphylococcus should be consistent throughout the text.

Antimicrobial resistance of isolated bacteria from HIV-infected women attending PMTCT clinic at Bukoba Municipality: Check Line 4 “….and 3(14.0%) for nitrofurantoin (Table 4)”. The correct table reference for nitrofurantoin is Table 3, not 4.

Factors associated with bacteriuria in HIV-infected women attending PMTCT clinic at Bukoba Municipality: Check the numbering of the tables-Table 6, not Table 2.

Formatting of Table 6 (wrongly titled as Table 2): • The table is incorrectly titled Table 2. Please correct the title to Table 6. The table exhibits inconsistent line and column formatting. Ensure that the text within the table follows a uniform layout for clarity and readability. The table suffers from inconsistent line/column formatting. The text in the table should follow a uniform layout.

DISCUSSION:

The authors should elaborate more on the multivariable modified Poisson regression analysis for bacteriuria (see last paragraph in the discussion section)

CONCLUSIONS:

The authors should include potential future directions for research.

Line 1: “E. coli” should be in italics

STUDY LIMITATIONS: Participants may experience recall bias. If they were unable to remember all relevant information during data collection, this limitation should be acknowledged in the discussion section.

RECOMMENDATIONS: I recommend merging the recommendations with the conclusions section to create a more cohesive final study summary. Also, correct “National Essential Medicines List (STG/NEMLIT) to avoid rational antimicrobial therapies.” Check the usage of “rational” in this context.

Availability of data and material: PLOS journals require authors to make all data necessary to replicate their study’s findings publicly available without restriction at the time of publication. Stating ‘data available upon request from the corresponding author’ is insufficient.

Supplementary materials: The authors should include pictures of the media plates used in the study. Additionally, data measuring the zones of inhibition for the antimicrobial susceptibility testing (AST) should be provided.

Reference: The title of the research article in Reference 2 should be included for completeness.

For Reference 7, please ensure the author's name is formatted in sentence case to comply with the citation guidelines.

**Reviewer #4:**  Thank you for the opportunity to review this manuscript on Aetiology, antimicrobial susceptibility patterns and factors associated with bacteriuria among HIV-infected women attending prevention of mother to child transmission clinic at Bukoba Municipality, Tanzania.

The time and effort put into this work is acknowledged and greatly appreciated.

The following are my comments and revision suggestions to the authors to improve the quality of their submissions.

• The title should be “Aetiology, antimicrobial susceptibility patterns and factors associated with bacteriuria among HIV-infected women attending the Prevention of Mother-to-Child Transmission (PMTCT) clinic at Bukoba Municipality, Tanzania.”

• Did you want to study asymptomatic bacteriuria, as defined in your introduction, or symptomatic bacteriuria, which is UTI? You cannot define bacteriuria as “without the presence of symptoms” if there are two types.

• Pregnant women living with HIV belong to a defined population. Why did you choose them? What is your rationale? Are there any risks associated with bacteriuria (symptomatic or not) in them? Why is this of concern to them and public health in general?

• The title of your study does not include lactating women living with HIV. They are brought into the write-up somewhere in the middle and not mentioned in the discussion and conclusion. Were they important to this study in the first place? Why is bacteriuria (symptomatic or not) important in these women? What is the importance of lactation to bacteriuria?

• Informed consent is built on the voluntary agreement of people to participate in a study after every relevant piece of information has been made available to them. Convincing them erodes their freedom to participate and deviates from the ethics of research involving humans. Did you explain the study in detail and allow them to choose to participate or did you convince them as you have written?

• Your discussion should be in shorter paragraphs, each addressing a specific finding. This will make it easier to follow your points.

• Your manuscript is titled “… and factors associated with bacteriuria among HIV-infected women …” but at the end of your discussion, you declared that no factors were associated with bacteriuria in this study. If you aimed to demonstrate the significance or otherwise of pre-defined factors to the incidence of bacteriuria in this population, then it would be acceptable to declare that you found no association between the variables but this is not the case here. In your results, you will find some factors associated with a higher incidence of bacteriuria in the population you studied. Discuss them. They do not necessarily have to be causative.

• I am sure that you want your study to be relevant to the care and management of pregnant women living with HIV. How does your study contribute to the prevention and control of bacteriuria in these women? You should include this in your conclusion and recommendations.

• Your recommendations address the judicious use of antibiotics but is that all there is to prevent and control bacteriuria in this population if it is important enough to be studied?

Good job, authors!

6. PLOS authors have the option to publish the peer review history of their article (what does this mean? ). If published, this will include your full peer review and any attached files.

**Do you want your identity to be public for this peer review?** For information about this choice, including consent withdrawal, please see our Privacy Policy .

Reviewer #1: No

Reviewer #2: No

Reviewer #3: **Yes: ** Damilola Odumade

Reviewer #4: **Yes: ** Dr. Oluwafolayemi Doyeni

---

## [Author Response · Author response to Decision Letter 1]

20 Jan 2025

RESPONSE TO REVIEWER’S COMMENTS

Title: Aetiology, antimicrobial Susceptibility Patterns and factors associated with bacteriuria among HIV-infected women attending Prevention of Mother to Child Transmission Clinic at Bukoba Municipality, Tanzania

Editor Comments

Thank you for stating the following financial disclosure: We received a small fund for data collection from TFELTP. Please state what role the funders took in the study. If the funders had no role, please state: "The funders had no role in study design, data collection and analysis, decision to publish, or preparation of the manuscript."

Response: The statement has been added to the cover letter to show that the funders had no role in study design, data collection and analysis, decision to publish, or manuscript preparation.

We note that your Data Availability Statement is as follows: All relevant data are within the manuscript and its Supporting Information files. Please confirm at this time whether or not your submission contains all raw data required to replicate the results of your study. Authors must share the “minimal data set” for their submission. PLOS defines the minimal data set to consist of the data required to replicate all study findings reported in the article, as well as related metadata and methods (https://journals.plos.org/plosone/s/data-availability#loc-minimal-data-set-definition).

Response: - We confirm that the submission contains all raw data required to replicate the result of the study

When completing the data availability statement of the submission form, you indicated that you will make your data available on acceptance. We strongly recommend all authors decide on a data sharing plan before acceptance, as the process can be lengthy and hold up publication timelines. Please note that, though access restrictions are acceptable now, your entire data will need to be made freely accessible if your manuscript is accepted for publication. This policy applies to all data except where public deposition would breach compliance with the protocol approved by your research ethics board. If you are unable to adhere to our open data policy, please kindly revise your statement to explain your reasoning and we will seek the editor's input on an exemption. Please be assured that, once you have provided your new statement, the assessment of your exemption will not hold up the peer review process.

Response: The data is available and agree to provide the necessary information whenever needed, but if the exemption is accepted I will appreciate it.

PLOS requires an ORCID iD for the corresponding author in Editorial Manager on papers submitted after December 6th, 2016. Please ensure that you have an ORCID iD and that it is validated in Editorial Manager. To do this, go to ‘Update my Information’ (in the upper left-hand corner of the main menu), and click on the Fetch/Validate link next to the ORCID field. This will take you to the ORCID site and allow you to create a new iD or authenticate a pre-existing iD in Editorial Manager.

Response: The ORCID ID has been updated on my account. The ORCID ID is 0000-0002-5793-2867

One of the noted authors is a group or consortium. In addition to naming the author group, please list the individual authors and affiliations within this group in the acknowledgments section of your manuscript. Please also indicate clearly a lead author for this group along with a contact email address.

Response: The acknowledgment section and the leading author have been indicated together with a contact email address.

Please amend either the abstract on the online submission form (via Edit Submission) or the abstract in the manuscript so that they are identical.

Response: The abstract on the online submission form has been amended to align with the abstract in the manuscript so that they are identical

Response: The reference has been revised.

Additional Editor Comments

When submitting the revised manuscript, please ensure that line numbers and page numbers are included for ease of reference in the review process

Response: Line numbers and page numbers have been added.

In the Introduction section, please define PMTCT upon its first mention and clarify any other abbreviations that have not been previously defined

Response: We have defined PMTCT and other abbreviations upon their first mention Page 4, L110

In the discussion section, please provide an interpretation of your study results, discussing the implications of these findings in the context of similar studies in the literature. Avoid repeating results in this section.

Response: The discussion section has been revised to accommodate the suggestion. Page ---, lines -----

Remove the sub-headings "Limitations" and "Recommendations." The limitations of the study should be detailed in the final paragraph of the Discussion section, while recommendations based on your findings should appear as the last paragraph of the Conclusion section.

Response: Subheadings for limitations and recommendations have been removed and the last paragraph of the discussion and conclusion revised accordingly. Page 17 L448-468.

Ensure that bacterial names are consistently italicized throughout the manuscript. Specifically, please italicize E. coli in the Conclusion section.

Response: We have reviewed the entire manuscript and consistently italicized the bacterial names including E. coli in the conclusion section. Pg 17, L455

REVIEWER #1

The study offers valuable insights into the high prevalence of bacteriuria and alarming levels of antibiotic resistance among HIV-infected women in a resource-limited setting. The findings align with regional studies; however, the absence of significant associations with bacteriuria risk factors indicates a need for more comprehensive investigations to identify potential predictors and effectively target interventions. The identification of a high proportion of multidrug-resistant (MDR) pathogens (68.2%) raises concern. It underscores the urgent need for improved antimicrobial stewardship, rational antibiotic use, and enhanced infection control measures to combat antibiotic resistance in vulnerable populations.

Response: Thank you for your comment; we have taken some comments as recommendations for further study.

The data and analyses robustly support the claims, and the described methodology is sufficient for reproducing the experiments. The authors appropriately contextualize their claims within the existing literature, fairly treating previous studies. They compare the prevalence of bacteriuria and MDR pathogens in their research with findings from similar studies conducted in Iran, Ethiopia, Nepal, Nigeria, and other regions of Tanzania, acknowledging geographical, methodological, and healthcare delivery differences that may explain variations in results

Response: Thank you for the comments

While the study effectively highlights high resistance rates linked to irrational use and easy access, incorporating clinical data on antibiotic use within and outside the PMTCT clinic could further enhance the analysis. This addition would facilitate correlations between resistance patterns and factors such as age, socioeconomic status, or prior antimicrobial use, leading to a more nuanced understanding of AMR drivers and improving the study's capacity to inform targeted interventions.

Response: This is well noted and appreciated, but during data collection, we didn’t collect data outside the PMTCT clinic, so it was hard to conduct analysis and make comparisons.

The authors' recommendations serve as a good starting point but could be strengthened for greater impact. Emphasizing the importance of continuous education for healthcare providers on antibiotic stewardship would enhance their recommendations. Additionally, defining rational use in the context of local resistance patterns and incorporating strategies for infection prevention and control would create a more comprehensive approach to addressing antimicrobial resistance.

Response: We have revised the recommendations emphasizing a need for education to healthcare providers on antibiotic stewardship, IPC, and rational use of antibiotics. Page 19, L463-466

The write-up could also benefit from minor edits to correct organization and typographical errors for improved clarity.

Response: The entire manuscript has been reviewed for typographical/grammatical errors.

REVIEWER #2

In the title, instead of “prevention of mother to child transmission” capitalize the first words of the clinic name “Prevention of Mother to Child Transmission (PMTCT)”.

Response: Thank you for the comment, the first words have been capitalized as suggested. page 1, L1-3

While the ethical considerations are well addressed, including information about how confidentiality was maintained during the study would strengthen this section.

Response: The information about the confidentiality of participants has been added, to page 8 L246-250

REVIEWER #3

The in-text citations are inconsistent (a mix of round and square brackets). Please note that PLOS One requires the use of reference numbers in square brackets, so this needs to be corrected throughout the manuscript.

Response: The square brackets have been used throughout the manuscript in reference citation.

Include page numbers and line numbers in the submitted manuscript for easier reference.

Response: Page and line numbers have been included in the manuscript

Ensure that the spacing between paragraphs is consistent across the document.

Response: We have checked the whole document to ensure that the spacing between paragraphs is consistent.

Under the “Financial Disclosure” section, please write out the full name of the funding body, specifically for TFELTP.

Response: The full name of the funding body TFELTP has been written in the financial disclosure section

The corresponding author list should not include physical addresses; only email addresses are required. List the corresponding author’s initials in parentheses after the email address. (See PLOS One Title, Author, Affiliations Formatting Guidelines)

Response: The physical address of the corresponding author has been removed, and the list of authors has been written according to PLOS ONE guidelines. Page 1

Abstract

1. In the sentence “17 (34.0%) of gram-negative bacteria were Extended Spectrum Beta Lactamase (ESBL) procedures,” please clarify if "procedures" is the intended term, as it seems out of context. Consider revising for clarity.

Response: The intended word was “producers”. The statement has been revised for clarity. Page1, L41.

2. In the phrase “Escherichia coli showed high rate of resistance rate against trimethoprim-sulfamethoxazole 21(100%),” please choose one term for consistency. For example, use "high rate of resistance" or "resistance rate" but not both.

Response: The phrase has been revised by choosing “high rate of resistance” for consistency. Page 2, L42

3. The statement under Recommendations is vague. The authors should specify whether they are referring to prescribed or over-the-counter antimicrobial agents.

Response: The recommendations have been revised for clarity. Page 2, L50-51

Introduction

Paragraph 3: Please state the country in which Mwanza is located for clarity.

Response: The country (Tanzania) where Mwanza is located has been added. Page 3, L73

Paragraph 4: Ensure that genus/species names are written in italics to adhere to scientific formatting conventions.

Response: Genus/species name has been written in italics to adhere to scientific formatting. Page 3

Material And Methods

Urine specimen collection and handling: 10 ml not ten mL

Response: Corrections has been made. Page 5, L145

Reference 28 is incorrectly cited. This study examined the effect of handwashing with water or soap and did not involve the use of a sterile cotton swab soaked in normal saline to reduce contamination. Therefore, this reference should be placed after "soap" for accurate attribution.

Response: We have corrected the position of reference 28, as suggested. Page 5, L148

Urine culture and bacteria identification: It would be great if the authors could state why a hydrogen sulfide production test was performed when the Kligler Iron Agar tests for hydrogen sulfide production.

Response: According to the laboratory SoP, both SIM and KIA tests are used to identify gram-negative bacteria; hydrogen sulfide is tested on methods. Page 6, L162-165

Results

Socio-demographic and clinical characteristics of the study participants: There is a lack of agreement between the data in Table 2 and the accompanying text (See the last line under this section of the manuscript).

Response: We have noted the lack of agreement between the data in the table and text. We have corrected the table. Page 8-9, L255-267

Table 3: Isolates “genus and species name” should be in italics.

Response: The genus and species name in Table 3 have been written in italics. Page 11,

The letter case for the acronym for Coagulase Negative Staphylococcus should be consistent throughout the text.

Response: The acronym for Coagulase Negative Staphylococcus has been revised to be consistent throughout the text.

Antimicrobial resistance of isolated bacteria from HIV-infected women attending PMTCT clinic at Bukoba Municipality: Check Line 4 “….and 3(14.0%) for nitrofurantoin (Table 4)”. The correct table reference for nitrofurantoin is Table 3, not 4.

Response: Thanks for the comment. This has been updated from table 4 to table 3. Page 11

Factors associated with bacteriuria in HIV-infected women attending PMTCT clinic at Bukoba Municipality: Check the numbering of the tables-Table 6, not Table 2

Response: The table number has been revised from table 2 to table 6.

Formatting of Table 6 (wrongly titled as Table 2): The table is incorrectly titled Table 2. Please correct the title to Table 6. The table exhibits inconsistent line and column formatting. Ensure that the text within the table follows a uniform layout for clarity and readability. The table suffers from inconsistent line/column formatting. The text in the table should follow a uniform layout.

Response: We corrected the table number to Table 6 and formatted the lines and columns to exhibit consistency. Pages 16 and 17.

Discussion

The authors should elaborate more on the multivariable modified Poisson regression analysis for bacteriuria (see last paragraph in the discussion section)

Response: We have added the description to elaborate more on the multivariable modified Poisson regression analysis for bacteriuria. Page 16, L439-445

Conclusions:

The authors should include potential future directions for research.

Response: Potential future directions for research have been added. Page 17, L458

Line 1: “E. coli” should be in italics

Response: E. coli in conclusion has been written in italics form. Page 17, L455

Study Limitations

Participants may experience recall bias. If they were unable to remember all relevant information during data collection, this limitation should be acknowledged in the discussion section.

Response: The limitation of recall bias has been acknowledged in the discussion section. Page 16, L150

Recommendations

I recommend merging the recommendations with the conclusions section to create a more cohesive final study summary. Also, correct the “National Essential Medicines List (STG/NEMLIT) to avoid rational antimicrobial therapies.” Check the usage of “rational” in this context.

Response: Recommendations have been merged with conclusions, and the word rational has been revised. Page 17, L461-468

Availability of data and material

PLOS journals re

---

## [Decision Letter · Decision Letter 1]

4 Feb 2025

Aetiology, antimicrobial susceptibility patterns and factors associated with bacteriuria among HIV-infected women attending prevention of mother to child transmission clinic at Bukoba Municipality, Tanzania

PONE-D-24-33570R1

Dear Dr. Felician,

We’re pleased to inform you that your manuscript has been judged scientifically suitable for publication and will be formally accepted for publication once it meets all outstanding technical requirements.

Kind regards,

Mabel Kamweli Aworh, DVM, MPH, PhD. FCVSN

Academic Editor

PLOS ONE

Additional Editor Comments (optional):

Reviewers' comments:

Reviewer's Responses to Questions

**Comments to the Author**

1. If the authors have adequately addressed your comments raised in a previous round of review and you feel that this manuscript is now acceptable for publication, you may indicate that here to bypass the “Comments to the Author” section, enter your conflict of interest statement in the “Confidential to Editor” section, and submit your "Accept" recommendation.

Reviewer #3: All comments have been addressed

Reviewer #4: All comments have been addressed

2. Is the manuscript technically sound, and do the data support the conclusions?

Reviewer #3: Yes

Reviewer #4: (No Response)

3. Has the statistical analysis been performed appropriately and rigorously? 

Reviewer #3: Yes

Reviewer #4: (No Response)

4. Have the authors made all data underlying the findings in their manuscript fully available?

Reviewer #3: Yes

Reviewer #4: (No Response)

5. Is the manuscript presented in an intelligible fashion and written in standard English?

Reviewer #3: Yes

Reviewer #4: (No Response)

6. Review Comments to the Author

Reviewer #3: (No Response)

Reviewer #4: (No Response)

7. PLOS authors have the option to publish the peer review history of their article (what does this mean? ). If published, this will include your full peer review and any attached files.

**Do you want your identity to be public for this peer review?** For information about this choice, including consent withdrawal, please see our Privacy Policy .

Reviewer #3: **Yes: ** Damilola Odumade

Reviewer #4: **Yes: ** Dr. Oluwafolayemi Doyeni

---

## [Editor Report · Acceptance letter]

PONE-D-24-33570R1

PLOS ONE

Dear Dr. Felician,

I'm pleased to inform you that your manuscript has been deemed suitable for publication in PLOS ONE. Congratulations! Your manuscript is now being handed over to our production team.

Kind regards,

on behalf of

Dr. Mabel Kamweli Aworh

Academic Editor

PLOS ONE